# Impact of the WHO Safe Childbirth Checklist on safety culture among health workers: A randomized controlled trial in Aceh, Indonesia

**Lennart Kaplan**[1,2‡], **Katharina Richert**[3‡], **Vivien Hülsen**[1‡], **Farah Diba**[4],
**Marthoenis Marthoenis**[4], **Muhsin Muhsin**[4], **Samadi Samadi**[4], **Suryane Susanti**[4],
**Hizir Sofyan**[4], **Ichsan Ichsan**[4]*, **Sebastian Vollmer**[1]*

**1** University of Goettingen, Göttingen, Germany, **2** German Institute of Development and Sustainability, Bonn, Germany, **3** University of Mannheim, Mannheim, Germany, **4** Universitas Syiah Kuala, Banda Aceh, Indonesia

‡ LK, KR and VH contributed equally to this work as joint first authors.
* ichsan@unsyiah.ac.id (II); svollmer@uni-goettingen.de (SV)

## Abstract

The World Health Organization (WHO) developed the Safe Childbirth Checklist (SCC) to increase the application of essential birth practices to ultimately reduce perinatal and maternal deaths. We study the effects of the SCC on health workers safety culture, in the framework of a cluster-randomized controlled trial (16 treatment facilities/16 control facilities). We introduced the SCC in combination with a medium intensity coaching in health facilities which already offered at minimum basic emergency obstetric and newborn care (BEMonC). We assess the effects of using the SCC on 14 outcome variables measuring self-perceived information access, information transmission, frequency of errors, workload and access to resources at the facility level. We apply Ordinary Least Square regressions to identify an Intention to Treat Effect (ITT) and Instrumental Variable regressions to determine a Complier Average Causal Effect (CACE). The results suggest that the treatment significantly improved self-assessed attitudes regarding the probability of calling attention to problems with patient care (ITT 0.6945 standard deviations) and the frequency of errors in times of excessive workload (ITT -0.6318 standard deviations). Moreover, self-assessed resource access increased (ITT 0.6150 standard deviations). The other eleven outcomes were unaffected. The findings suggest that checklists can contribute to an improvement in some dimensions of safety culture among health workers. However, the complier analysis also highlights that achieving adherence remains a key challenge to make checklists effective.

**Data Availability Statement:** All relevant data are within the paper and its Supporting Information files.

## Introduction

The Lancet Global Health Commission on High-quality Health Systems suggests that in many settings unskilled labor or avoidable mistakes lead to a suboptimal level of care despite increasing access to the health system [1]. Particularly, childbirth remains a context that is likely to benefit from an improvement of health workers' attitudes and perceptions (which we subsume

**Funding:** 1. Lower Saxony Ministry of Science and Culture: Reducing Poverty Risk in Developing Countries (received by SV) 2. European Commission: Experts for Asia Scholarship Program. (received by KR and LK) 3. German Research Foundation (DFG): Research training group 1723 "Globalization and Development" (received by SV) The funders had no role in study design, data collection and analysis, decision to publish, or preparation of the manuscript.

**Competing interests:** The authors have declared that no competing interests exist.

in the following under the term safety culture). In 2015, approximately 303,000 maternal deaths occurred, and in 2016, 5.6 million children below the age of five died worldwide due to largely preventable causes [2, 3]. While the global neonatal mortality rate equaled 18.0 per 1,000 live births in 2017, the UN Sustainable Development Goal 3 aims to reduce the rate to 12.0 per 1,000 live births.

Checklists have been identified as a low-cost intervention to improve quality in different sectors [4, 5] and to contribute to a "safety culture" [6]. The World Health Organization (WHO) introduced a global collaborative effort to create a Safe Childbirth Checklist (SCC) [7]. The checklist consists of four pages, reminding users about essential practices to follow and includes a brief explanation for each point.

To assess the SCC's effectiveness, the WHO invited practitioners and academics alike to evaluate the checklist in different contexts. A systematic review of this evidence indicates that the SCC can increase essential birth practices, and may even reduce stillbirth [8]. Moreover, Walker et al. [9] provide causal evidence that the SCC reduced neonatal mortality in Kenya and Uganda when coupled with other interventions. However, above initial uptake, a change in safety culture is required to achieve sustainable behavioral change. Active usage of the SCC by health workers is a pre-condition for reaping those benefits, where, in several contexts, limited usage constrained the application of the checklist [10]. Observational studies conducted in Sri Lanka and Namibia, observed SCC usage rates of 45.8% and 75% during birthing processes [11, 12]. So far, there is only scant evidence from non-randomized studies on how the introduction of the SCC affected safety culture. Qualitative interviews among Brazilian nurses suggest that the checklist was recognized as a quality improvement tool [13] and changed routines, but also led to resistance [14]. A pre-post study conducted in Brazil concludes that improvements in adherence to essential birth practices and subsequently improved clinical outcomes may have been stimulated by context-specific factors such as teamwork, supportive leadership, quality improvement climate and indicator-based adjustments [15]. These elements, alongside open communication concerning errors, and appropriate workload are considered essential in fostering a safety culture [16], but up to this point a causal assessment of the SCC on health workers' perceptions is missing.

This study aims to address this gap by providing evidence from a cluster-randomized controlled trial on the SCC's impact on safety culture. We estimate treatment effects on 14 variables measuring the following dimensions of safety culture at the facility (cluster) level: self-perceived information access, information transmission, frequency of errors, workload and access to resources.

Additionally, to carve out the causal pathways, we engage into a complier analysis, which focuses on those midwives that adhere to the checklist. In other SCC evaluations, compliance was measured either in terms of completing single checklist items, certain bundles (e.g., before/after birth) or the full checklists over the total number of checklists provided [17–19], over the total number of live births [20], or over mothers admitted [15]. Other studies simply asked health care providers whether they carried out essential birth practices outlined in the SCC [21] (Abawollo et al. 2021).

## Materials and methods

### Ethics

Both the ethical board of the Universitas Syiah Kuala, Indonesia, (08/KE/FK/2016) and the ethics committee of the University of Göttingen, Germany (June 27, 2016) reviewed the study protocol and declared no objection prior to the trial. The research team registered the study as ISRCTN11041580 and AEARCTR-0003548. The study was supported by the Aceh provincial

health office and the district health offices of Aceh Besar, Banda Aceh, and Bireuen. We collected informed consent both from facility leadership and the interviewed midwives.

## Trial setting and participants

With a neonatal mortality rate of 34 deaths per 1000 live births, Aceh ranked 19th out of 34 provinces in Indonesia [22]. Starting in July 2016, the checklist was implemented in combination with a medium-intensity coaching of health workers in the districts of Aceh Besar, Banda Aceh, and Bireuen. We chose a clustered design at the facility level to offer a team-based light-touch implementation approach, to address the difficulty that births can take place across different shifts and to avoid spillover effects. Facilities were eligible if they offered at least basic emergency obstetric and newborn care services (BEMonC covers provision of oxytocin and antibiotics, manual removal of placenta, assisted vaginal delivery, removal of retained products, resuscitation of the newborn and monitoring of the newborn). BEMonC status was assessed via a survey with the head of the delivery room (obstetrician or senior midwife). Out of 40 eligible facilities, eight did not participate in the study as they either did not have any births during the past month or asked for financial compensation, which we could not provide. The sampled facilities (Fig 1) include hospitals, community health centers (puskesmas), and midwife clinics. Those facilities have a total yearly volume of around 11,000 births, which served as a threshold sample size from power calculations for health outcomes in our accompanying paper [23]. However, we also assessed minimal detectable effect sizes for our safety culture measures at baseline (see Table C in S1 Text).

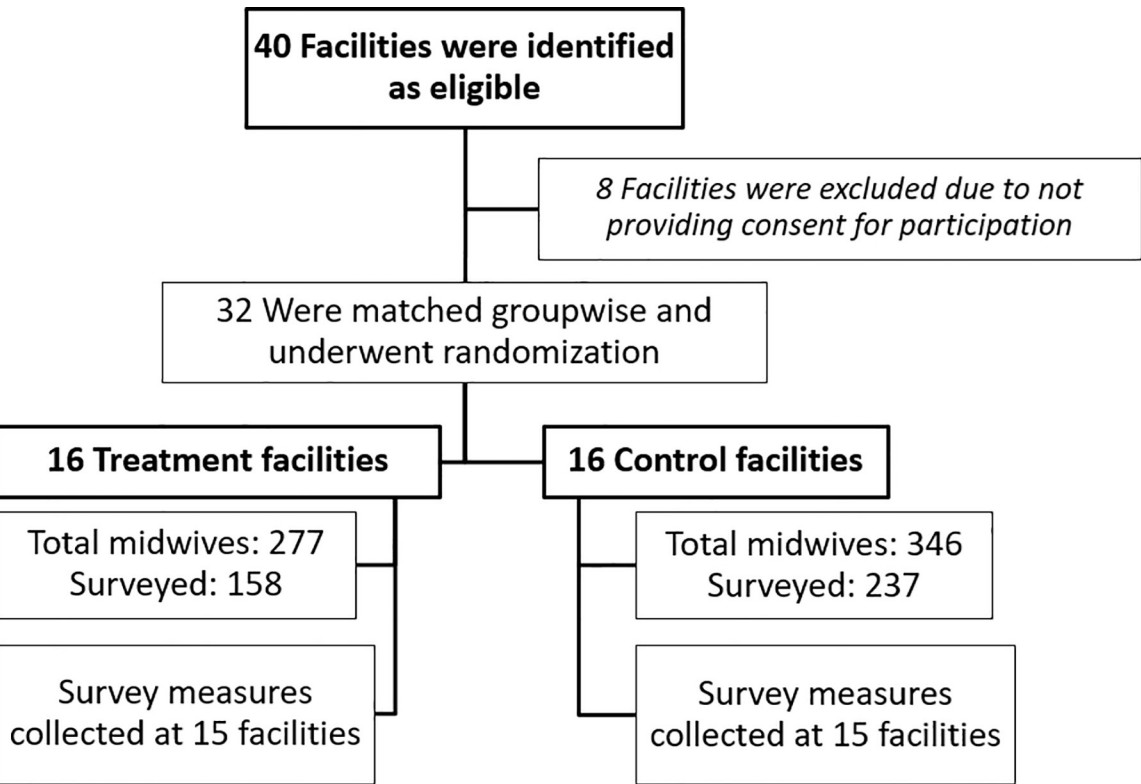

**Fig 1. Randomization and data collection.** Note: Authors' own depiction of randomization process.

## Intervention

In collaboration with local midwives, obstetricians, and policymakers, we adapted the checklist to the Acehnese context (See Fig A in S1 Text for the adapted checklist and the corresponding section for an overview about adaptations). Following the Engage–Launch–Support–Model of the accompanying Better Birth Trial in India [5], a team of coaches introduced the checklist in 16 health facilities in Indonesia. Since the coaches were responsible for multiple facilities, the checklist introduction was spread over several days in October 2016. The trained coaches provided motivating arguments and information on correct checklist use during a two-hour launch event with midwives and subsequently introduced the SCC. No technical training was provided given that the SCC is a reminder of essential practices that are part of the midwives' professional education (on average, midwives answered eight out of nine checklist-related knowledge questions correctly). Since the checklist takes a reminder function, coaches trained midwives to fill out checklist copies during the birth process. Additionally, our team provided a checklist poster for the delivery room. In emergencies or when not working in teams (e.g., during night shifts), midwives may have only filled out the paper checklist ex post. Over six months, the coaches visited each health facility eleven times to provide new checklist copies, collect filled-out SCCs, and provide feedback. The coaching was supported by two meetings with the Checklist Quality Coordinator (CQC), a responsible lead person chosen within the facility, to exchange best practices (the study protocols and the S1 Text provide a more detailed description of the intervention).

## Trial design/Randomization and masking

The research team used an optimization approach to match facilities in two groups [24]. Following this matching approach, we minimize the mean squared error between the two groups on covariates and potential outcomes before treatment. Table A in S1 Text indicates that the primary health outcomes and facility characteristics (i.e., facility/health organization types, offered services, sampled districts) that are likely to be correlated with individual safety culture were balanced at baseline, indicating a successful matching. After allocating two equally sized groups of 16 facilities, the research team assigned treatment and control status randomly by coin toss; and introduced the treatment subsequently in 16 facilities (50% of the sample) at the facility level. Enumerators collected also data in the 16 facilities of the control group to compare effects of the intervention. Blinding of study participants was not possible.

## Outcomes and data collection

We identify four dimensions of safety culture through which the adoption of the checklist could support adherence to essential practices during childbirth. First, by ticking the boxes and adding supplementary data, *information transmission* may be improved for midwives working in teams and shifts [25]. Moreover, the checklist may be an empowering point of reference to speak up against procedural flaws [4, 26] and to address the global issue of underreporting of mistakes in maternity care [27]. This way, the checklist may improve overall *information accessibility*. Information transmission and access hence refer to appropriate communication of relevant information and issues related to patient care, working in a well-coordinated team, access to information on patient medication and diagnosis. Second, the checklist is designed as a powerful reminder to recall essential steps during periods of high workload and stress like emergencies [28]. Thus, it may reduce the (perceived) *frequency of errors* and, this way, may improve the quality of care. Frequency of errors is measured as errors in relation to fatigue, distractions, knowledge and excessive workload. Third, as a job-aid, the checklist aims to reduce cognitive load. Nonetheless, the SCC creates further paperwork and

may reduce health workers' motivation [11]. Its implications on the (perceived) *workload* are, thus, ambiguous. We measure perceived workload as the burden of paperwork and a rating of the average workload within the facility. Fourth, the checklist effectiveness crucially depends on the availability of complementary supplies [29]. While our intervention did not include a provision of *supplies*, our coaching approach explicitly made midwives aware of insufficient supplies and stressed the requirement to demand supplies from district health offices. We investigate resource access through inquiring about the adequacy of tools and resources to effectively perform duties. For an overview of the survey items please consult Table F in S1 Text. Given the team-based intervention, we consider our main measures of interest on perceptions and attitudes for these four dimensions at the cluster level.

In addition to the checklist intervention, our team collected general information on health facilities and patient outcomes at the facility level and health worker characteristics and safety culture dimensions at the individual level. The individual questionnaires covered 376 (out of 623) midwives. We translated all items back and forth and determined the questionnaires' context sensitivity via a pre-test. Our main analysis is based on the post-intervention data collection in April 2017 with data on treatment and control clusters, while we also collected baseline information between August and October 2016 (e.g., for matching facilities). Trained enumerators collected the data randomly during the shifts of midwives via Computer-Assisted-Personal-Interviews using Question Pro software. The team recorded the data with anonymized IDs on the spot. The data was uploaded to a secure server. Data quality was ensured through high frequency checks, which specifically consisted of response quality and enumerator checks. On random occasions, enumerators were accompanied by the survey team to monitor the interview process. We build on those data to consider our primary outcomes as outlined in our trial registration at ISRCTN11041580. Despite the high baseline levels of certain outcomes and a potential ceiling effect, we capture sufficient variation to estimate tangible effects (see Table C in S1 Text). We lost one control facility and one treatment facility, in both cases because they did not have any births during the observation period. As we did not apply a pair-wise matching, this attrition did not further affect our analysis.

## Statistical analysis

We estimate Intention to Treat (ITT) effects using ordinary least squares regressions. The basic estimation equation for the Intention to Treat (ITT) effect reads as follows:

$$\bar{Y}_i = \alpha + \beta_1 T_i + \beta_2 X_i + \varepsilon_I$$

We consider measures of safety culture in the four dimensions information accessibility and transmission, perceived frequency of errors, workload, and resource access as primary outcomes $Y_i$. $T_i$ indicates the facility's treatment status refers to the team-based intervention at the facility level; hence, the outcome is the average safety culture at the facility level. $X_i$ is a vector of covariates (facility type, urban-rural, CeMonC-status, district dummy). While we do not need to include covariates in the regression due to the randomized study design, we report results with covariates to increase statistical efficiency in Table D in S1 Text. $\varepsilon_i$ is the error term.

Additionally, we estimate Complier Average Causal Effects (CACE) for the individuals who took up the treatment [30]. The CACE estimator predicts compliance with the treatment indicator in a first stage and consecutively estimates the treatment effect among the compliers in a second stage (for a more elaborate description of the approach, please see the S1 Text). We defined facility's degree of compliance as a continuous indicator based on the number of completed checklists over the total numbers of births. Across intervention facilities, midwives used the SCC for 48.6% of births. We estimated both CACE and ITT effects using Stata version 16.

## Results

### Changes in safety culture

Table 1 provides the treatment effects on health workers' safety culture and Fig 2 provides a coefficient plot, in which whiskers indicate a 10% confidence interval. Results on information access indicate neither significantly better diagnostic nor medication information. Regarding information transmission, the checklist does not improve coordination and has neither statistically significant effects on general communication flows nor on reporting. However, both for the ITT and the CACE, the results show a significant increase in self-reported ease of speaking up when noticing patient care issues (row 4). In line with the checklist's reminder function, rows 9–12 indicate a reduction in the frequency of errors, which are, however, only significant

**Table 1. Health worker safety culture—Treatment effects.**

| | Mean Control | Mean Intervention | ITT (in standard deviations) | CACE (in standard deviations) |
|---|---|---|---|---|
| **Information Accessibility and** | | | | |
| (1) **InfoAccess1:** During your shift, do you always have access to the following patient information: Diagnosis (Scale 1 [No access at all] to 4 [Full Access]) | 3.6689 | 3.6605 | -0.0282 [-0.789–0.733] | -0.0580 [-1.505–1.389] |
| (2) **InfoAccess2:** During your shift, do you always have access to the following patient information: Medication (Scale 1 [No access at all] to 4 [Full Access]) | 3.6271 | 3.670 | 0.1375 [-0.622–0.897] | 0.2832 [-1.170–1.736] |
| **Information Transmission** | | | | |
| (3) **InformationFlow:** Relevant information is communicated appropriately within the delivery team. (Scale 1 [Disagree strongly] to 6 [Agree strongly]) | 5.099 | 5.073 | -0.1240 [-0.884–0.636] | -0.2553 [-1.712–1.201] |
| (4) **SpeakUp:** In this clinical area, it is easy to speak up if I perceive a problem with patient care. (Scale 1 [Disagree strongly] to 6 [Agree strongly]) | 5.0194 | 5.1658 | 0.6945* [-0.018–1.407] | 1.4301** [0.046–2.814] |
| (5) **Coordination:** The delivery staff members here work together as a well-coordinated team. (Scale 1 [Disagree strongly] to 6 [Agree strongly]) | 5.1794 | 5.2204 | 0.1563 [-0.602–0.915] | 0.3219 [-1.101–1.745] |
| (6) **FreqMissed:** During your most recent delivery shift-week, how often did you forget to transmit important information during sign-out? (Scale 1 [Never] to 6 [Very often]) | 1.5545 | 1.5620 | 0.0203 [-0.741–0.781] | 0.0418 [-1.405–1.489] |
| (7) **FreqUnsure:** During your most recent delivery shift-week, how often did you report information that you were unsure of? (Scale 1 [Never] to 6 [Very often]) | 1.4092 | 1.5920 | 0.5123 [-0.223–1.247] | 1.0548 [-0.285–2.395] |
| Frequency of errors | | | | |
| (8) **ErrorKnow:** During your most recent delivery shift-week, how often did you make errors because of inadequate knowledge? (Scale 1 [Never] to 6 [Very often]) | 1.7843 | 1.5548 | -0.3788 [-1.126–0.368] | -0.7800 [-2.260–0.699] |
| (9) **ErrorFatigue:** During your most recent delivery shift-week, how often did you make errors because of fatigue? (Scale 1 [Never] to 6 [Very often]) | 1.6131 | 1.5265 | -0.2307 [-0.987–0.525] | -0.4749 [-1.922–0.972] |
| (10) **ErrorDist:** During your most recent delivery shift-week, how often did you make errors because of distractions? (Scale 1 [Never] to 6 [Very often]) | 1.5067 | 1.3033 | -0.3477 [-1.097–0.401] | -0.7160 [-2.178–0.746] |
| (11) **ErrorExc:** During your most recent delivery shift-week, how often did you make errors because of excessive workload? (Scale 1 [Never] to 6 [Very often]) | 1.8372 | 1.5215 | -0.6318* [-1.353–0.089] | -1.3010* [-2.849–0.247] |
| **Workload** | | | | |
| (12) **Paperwork:** Paperwork takes too much time. (Scale 1 [Disagree strongly] to 6 [Agree strongly]) | 2.8177 | 2.8887 | 0.0856 [-0.675–0.846] | 0.1763 [-1.268–1.621] |
| (13) **Workload:** How would you rate the average workload among your delivery staff at your health facility? (Scale 1 [Very low] to 5 [Very high]) | 3.0249 | 2.8732 | -0.5679 [-1.297–0.161] | -1.1692 [-2.655–0.316] |
| **Access to resources** | | | | |
| (14) **ResAcc:** Do you have access to the tools and resources to do your job well? (Scale 1 [Never] to 4 [Always]) | 3.4184 | 3.6089 | 0.6150* [-0.108–1.338] | 1.2663* [-0.161–2.694] |
| Observations | 15 | 15 | 30 | 30 |

Notes: The dependent variable is depicted in standard deviations. 95% Confidence Intervals are depicted in brackets and significance levels are indicated by stars: p-val *<10%, **<5%, ***<1%. F-Statistics for the CACE estimator is for all regressions {39.639}. Cragg-Donald Wald F statistics>10 suggest that the treatment is a sufficiently strong predictor of compliance to warrant reliable inference (e.g., we do not face weak instrumental variable issues).

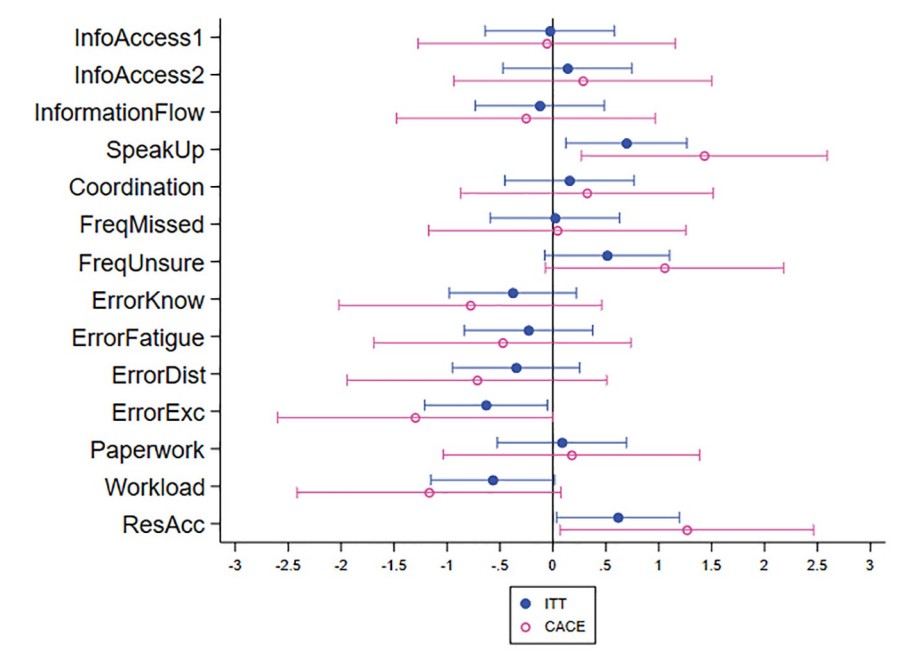

**Fig 2. Health worker safety culture.** Note: Estimates refer to changes in standard deviations. Table 1 provides the estimated coefficients.

for the reduction of perceived errors during periods of excessive workload (row 12). We also consider the effects on perceived workload. Yet, neither perceived paperwork nor general workload are affected. To keep implementation costs manageable, we deliberately decided to not link the intervention with a provision of additional supplies or training. Yet, our coaching provided input on where to access support if needed. Interestingly, there seems to be a perceived improvement in resource access for the intervention group. We also report results with basic covariates (facility type, urban-rural, CEmONC-status, district dummy) in Table D in S1 Text.

## Discussion

The literature stresses the need for measures to improve the quality of care to achieve the health-related SDGs, particularly in resource-constrained settings [1, 31]. Checklists appear as a particularly suitable tool to improve quality of care in such settings at a comparatively low cost. Previous studies indicate promising effects of the SCC on the application of essential birth practices [12, 23, 28] and a reduction of stillbirths and neonatal mortality in some contexts—e.g., in Kenya and Uganda [9] or in Rajasthan, India [32]. In contrast, no mortality reduction occurred in other settings [28], However, so far there was little evidence why the SCC works in one context, but not the other.

Previous trials suggest low uptake and a tapering out of effects as a potential barrier [23, 33], which is supported by limited qualitative evidence on health workers' resistance to the change in routines [14]. Thus, it is key to understand the perceptions of midwives to sustain checklist use in a more rigorous manner. This study examined in Aceh province, Indonesia, how the checklist affects perceptions on safety culture in the four dimensions information accessibility and transmission, perceived frequency of errors, workload, and resource access as

primary outcomes. The checklist seems to empower midwives to report issues with patient care, which was previously identified by qualitative work as an important issue in the context of Aceh [25]. While midwives reported in focus groups that paperwork would be a burden in their everyday work [23], the present study's results suggest that the checklist seems not to contribute significantly to a perceived increase in workload.

The checklist further reduces self-perceived errors during periods of high workload. Moreover, the checklist improves self-perceived availability of supplies, which points to empowering effects for midwives to demand necessary resources, given that we did not provide any resources ourselves [34]. Such positive effects of the SCC on resource access are particularly notable as supplies were a binding constraint for the effectiveness of the SCC in other settings [35]. The other dimensions of safety culture remained unaffected.

This study has some limitations. The sample is relatively small which limits statistical power. All outcomes are self-reported and self-perceived measures of providers aggregated at the facility level. Self-reported measures might be different, in either direction, from actual outcomes. Results are specific to the context of Aceh in Indonesia and lack of external validity to other settings might be an even larger concern for outcomes of safety culture than it is anyways for any RCT. Finally, we do not study the impact of safety culture on improved quality of care at the facilities.

Future research may want to validate whether the perceived improvements in errors during times of high workload and reporting of maltreatment also translates into actual practices, particularly if bundled with a modified checklist design or with more comprehensive implementation programs [9, 21, 36–38]. Research on safety culture from other implementation sites could help to further unravel the mixed results of SCC implementation across contexts and take user experiences of health staff into account to continuously improve the tool [39].

## Supporting information

**S1 Text. Supplementary appendix, which provides further information on the sampling, the intervention, the data collection and the statistical method.**
(DOCX)

## Acknowledgments

For their excellent research assistance and support of the study we thank Masyitah and Grit Försterling as well as the numerous local research assistants: Mutia Elviani, Amanda Putri Kairina, Aulia, Misna, Fitra Jaya Saputra, Hujjatul Balighah, Khairiah, Raziah, Nuriana, Nurul Fajar, Riska Alfiani, Teddy Kurniady Thaher, Fitriatul Ula, Ruzwar Wahyudi, Zahra Sofia, Zulfazli, Alfiyatul Rahmi, Cynthia Eka Putri, Fauziah. We also very much appreciated the thoughtful proofreading by Erin Flanagan and Laura Mahoney. We are incredibly grateful to the team from AriadneLabs, who trained our coaches and provided advise at any point during study design and implementation. Finally, we would like to thank Lisa Rogge, Stefan Klonner, Manuela de Allegri, Stephan Brenner, Christine Binzel, Simon Quinn, Anna Merkel, Christian König, Miriam Romero, Christoph Kubitza, Sebastian Prediger, Jenny Aker, David McKenzie, Sebastian O. Schneider, and Florina Serbanescu as well as two anonymous referees for their valuable comments.

## Author Contributions

**Conceptualization:** Lennart Kaplan, Katharina Richert, Farah Diba, Marthoenis Marthoenis, Hizir Sofyan, Sebastian Vollmer.

**Data curation:** Lennart Kaplan, Katharina Richert, Vivien Hülsen, Farah Diba, Marthoenis Marthoenis, Muhsin Muhsin, Samadi Samadi, Suryane Susanti, Ichsan Ichsan, Sebastian Vollmer.

**Formal analysis:** Lennart Kaplan, Katharina Richert, Vivien Hülsen.

**Funding acquisition:** Lennart Kaplan, Katharina Richert, Sebastian Vollmer.

**Investigation:** Lennart Kaplan, Katharina Richert, Sebastian Vollmer.

**Methodology:** Lennart Kaplan, Katharina Richert, Sebastian Vollmer.

**Project administration:** Farah Diba, Marthoenis Marthoenis, Muhsin Muhsin, Suryane Susanti, Hizir Sofyan, Ichsan Ichsan.

**Supervision:** Lennart Kaplan, Katharina Richert, Marthoenis Marthoenis, Samadi Samadi, Suryane Susanti, Ichsan Ichsan, Sebastian Vollmer.

**Writing – original draft:** Lennart Kaplan, Katharina Richert, Vivien Hülsen, Ichsan Ichsan, Sebastian Vollmer.

**Writing – review & editing:** Lennart Kaplan, Vivien Hülsen, Farah Diba, Marthoenis Marthoenis, Muhsin Muhsin, Samadi Samadi, Suryane Susanti, Hizir Sofyan, Sebastian Vollmer.

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
