## [Decision Letter · Decision Letter 0]

12 Dec 2022

PGPH-D-22-01265

Impact of the WHO Safe Childbirth Checklist on Safety Culture among Health Workers: A Randomized Controlled Trial in Aceh, Indonesia

Dear Dr. Vollmer,

Thank you for submitting your manuscript to PLOS Global Public Health. After careful consideration, we feel that it has merit but does not fully meet PLOS Global Public Health’s publication criteria as it currently stands. Therefore, we invite you to submit a revised version of the manuscript that addresses the points raised during the review process.

Constructive feedback from three reviewers is included below. As they highlight, further explanation of the intervention itself as well as trial methods are needed. Please ensure that reporting adheres to the appropriate CONSORT checklist and that all concerns raised by reviewers are fully addressed in the manuscript and response letter. 

We look forward to receiving your revised manuscript.

Kind regards,

Hannah Tappis, DrPH, MPH

Academic Editor

Journal Requirements:

2. Please send a completed 'Competing Interests' statement, including any COIs declared by your co-authors. If you have no competing interests to declare, please state "The authors have declared that no competing interests exist". Otherwise please declare all competing interests beginning with the statement "I have read the journal's policy and the authors of this manuscript have the following competing interests:"

3. Please amend your detailed Financial Disclosure statement. This is published with the article. It must therefore be completed in full sentences and contain the exact wording you wish to be published.

4. Please provide separate figure files in .tif or .eps format and remove the embedded figures from the manuscript file.

5. In the online submission form, you indicated that your data will be submitted to a repository upon acceptance.  We strongly recommend all authors deposit their data before acceptance, as the process can be lengthy and hold up publication timelines. Please note that, though access restrictions are acceptable now, your entire data will need to be made freely accessible if your manuscript is accepted for publication. This policy applies to all data except where public deposition would breach compliance with the protocol approved by your research ethics board. If you are unable to adhere to our open data policy, please kindly revise your statement to explain your reasoning and we will seek the editor's input on an exemption. Please be assured that, once you have provided your new statement, the assessment of your exemption will not hold up the peer review process.

Additional Editor Comments (if provided):

Reviewers' comments:

Reviewer's Responses to Questions

**Comments to the Author**

1. Does this manuscript meet PLOS Global Public Health’s publication criteria? Is the manuscript technically sound, and do the data support the conclusions? The manuscript must describe methodologically and ethically rigorous research with conclusions that are appropriately drawn based on the data presented.

Reviewer #1: Yes

Reviewer #2: Partly

Reviewer #3: Yes

2. Has the statistical analysis been performed appropriately and rigorously?

Reviewer #1: Yes

Reviewer #2: Yes

Reviewer #3: Yes

3. Have the authors made all data underlying the findings in their manuscript fully available (please refer to the Data Availability Statement at the start of the manuscript PDF file)?

Reviewer #1: No

Reviewer #2: No

Reviewer #3: Yes

4. Is the manuscript presented in an intelligible fashion and written in standard English?

Reviewer #1: Yes

Reviewer #2: Yes

Reviewer #3: Yes

5. Review Comments to the Author

Reviewer #1: This is an innovative study that measures the effects of using the WHO Safe Childbirth Checklist in the context of the safety culture of health services.

The article was analyzed according to CONSORT protocol criteria extension for cluster designs. The comments below indicate minor adjustments to be made to the text.

TITLE: It’s necessary to insert in the title the information that’s a cluster randomised trial study.

ABSTRACT:

Even knowing that there are word limits for the abstract, I strongly suggest that authors consider three aspects to be added:

1) the eligibility criteria of the clusters.

2) the number of clusters randomized to each group.

3) the primary outcomes (the four dimensions) as well as the information that primary outcome pertains to the cluster level.

INTRODUCTION

At the end of the introduction, the authors report the results of the study itself. I suggest rewriting the text putting the hypotheses related to the use of the checklist more comprehensively, considering the dimensions of the safety culture analyzed, without reporting your own results that should be presented in another section.

METHODS

It’s unclear whether the number of subjects interviewed at baseline and after six months was the same, or whether there were losses.

Why in Appendix Tables A2 and A3 there are no results for the criteria Information Accessibility (InfoAccess1 and InfoaAccess2).

In Appendix Table 2 it’s not clear p value refer <10%, <5% or <1%.

RESULTS:

It’s necessary to correct Figure 2. There is no correspondence with what is in table A4. For example, in Table A4 there are 14 safety culture criteria and in Figure there are 17. Furthermore, the text says that “the results show a significant increase in the probability that midwives would speak up when noticing patient care issues (row 4)”. However, in the figure, row 4 corresponds to "coordination”.

To understand the information in Figure 2, it is necessary to refer to Table A7 in the Appendix. For example: when analyzing the Figure, it is not clear what InfoAccess1 means. I suggest the authors describe the criteria that were analyzed in the text (on topic Outcomes and Data collection).

Reviewer #2: Review of manuscript PGPH-D-22-01265, entitled: “Impact of the WHO Safe Childbirth Checklist on Safety Culture among Health Workers: A Randomized Controlled Trial in Aceh, Indonesia”

Thank you very much for this opportunity to review this manuscript. Studies on safety culture and safety checklists are interesting and important topics to study, as the authors have made clear. However, I do have some questions to the manuscript.

The study is described as a cluster randomized controlled trial. It is a bit unclear if it actually is a cross-over design in the trial? In a stepped wedge cluster RCT, the clusters serve as their own controls and the intervention is administered to all cluster, and at the end of the trial all clusters have received it. In a cluster RCT there will be ‘pockets’ of controls in the intervention period, which does not seem to be the case in this trial? At least it is difficult to assess whether this is the case or not. In the consort extention checklist, there is a reference to parallel or factorial trial design. It would be helpful if the design could be more clarified in the manuscript.

Did all 16 facilities receive the intervention at the same time point or at different time points? Again, this might have implications for the design used in the trial.

Who performed the randomization? Please add a description of this.

Randomisation – fasilities from the same health organization, mix or different organizations allocated into treatment or controls?

Surveys were delivered on phone to participants. How was data privacy protection ensured for participants? How was data quality ensured? Please provide a brief description on this.

Paper checklist – how was it filled in/used? During the procedure, before or after? This could be described in more detail, to help the reader to understand the context of using the checklist.

Ethics. Do the authors have a reference number for ethical reviews from the ethical boards reviewing the study? Under ethics, it says that the study was reviewed, but not that it was approved. If it was approved prior to start, please mention this under this section.

How many respondents filled in the survey? How many nurses/midwifes responded? What does 15 observations represent? Facilities or nurses? Please clarify this.

Compliance – who filled in the surveys? Did the survey respondents attend to the teaching classes?

Results – changes in safety culture – midwifes would speak up, is stated in Results, for both ITT and CACE. In Table A4 and A5, only CACE has a significant confidence interval. This means that the results are not accurately reported, in results or in the tables. This need to be corrected.

Table A1 In the columns, what are the numbers (except * and **) representing? Mean scores or frequencies? This may be difficult to assess for the readers. Please clarify what all numbers represent. What are the scores for each variable in Table A1?

In Table A3 ,ICC is described under the table but it is not listed in the columns? What is the ICC values here? This needs to be more detailed.

The study include four dimensions, which the authors align with a safety culture. These are named: “information transmission”, “information accessibility”, “frequency of errors”, and “workload”. There is no data on the reliability of these dimensions. Please provide this for all four.

There are few questions with positive changes, hence the conclusion on safety culture being affected by the checklist may not seem completely justified based on the data.

Reviewer #3: Thank you for the opportunity to review this informative manuscript. There is a clear and compelling need to explore ways that may lead to culturally appropriate implementation of the Safe childbirth Checklist (SCC). The issue of perceptions toward safety practices among providers of facility delivery care is important and not well explored previously. Please see detailed comments in the attached file.

6. PLOS authors have the option to publish the peer review history of their article (what does this mean?). If published, this will include your full peer review and any attached files.

**Do you want your identity to be public for this peer review?** For information about this choice, including consent withdrawal, please see our Privacy Policy.

Reviewer #1: No

Reviewer #2: No

Reviewer #3: **Yes: **Florina Serbanescu

---

## [Decision Letter · Decision Letter 1]

8 May 2023

Impact of the WHO Safe Childbirth Checklist on Safety Culture among Health Workers: A Randomized Controlled Trial in Aceh, Indonesia

PGPH-D-22-01265R1

Dear Vollmer,

We are pleased to inform you that your manuscript 'Impact of the WHO Safe Childbirth Checklist on Safety Culture among Health Workers: A Randomized Controlled Trial in Aceh, Indonesia' has been provisionally accepted for publication in PLOS Global Public Health.

Best regards,

Hannah Tappis, DrPH, MPH

Academic Editor

Reviewer Comments (if any, and for reference):

Reviewer's Responses to Questions

**Comments to the Author**

1. If the authors have adequately addressed your comments raised in a previous round of review and you feel that this manuscript is now acceptable for publication, you may indicate that here to bypass the “Comments to the Author” section, enter your conflict of interest statement in the “Confidential to Editor” section, and submit your "Accept" recommendation.

Reviewer #3: All comments have been addressed

2. Does this manuscript meet PLOS Global Public Health’s publication criteria? Is the manuscript technically sound, and do the data support the conclusions? The manuscript must describe methodologically and ethically rigorous research with conclusions that are appropriately drawn based on the data presented.

Reviewer #3: Yes

3. Has the statistical analysis been performed appropriately and rigorously?

Reviewer #3: Yes

4. Have the authors made all data underlying the findings in their manuscript fully available (please refer to the Data Availability Statement at the start of the manuscript PDF file)?

Reviewer #3: Yes

5. Is the manuscript presented in an intelligible fashion and written in standard English?

Reviewer #3: Yes

6. Review Comments to the Author

Reviewer #3: thank you for addressing my comments. I appreciate the thoughtful efforts to respond to my concerns.

7. PLOS authors have the option to publish the peer review history of their article (what does this mean?). If published, this will include your full peer review and any attached files.

**Do you want your identity to be public for this peer review?** For information about this choice, including consent withdrawal, please see our Privacy Policy.

Reviewer #3: **Yes: **Florina Serbanescu
